# Flexible and accurate inference and learning for deep generative models

**Eszter Vértes**    **Maneesh Sahani**
Gatsby Computational Neuroscience Unit
University College London
London, W1T 4JG
{eszter, maneesh}@gatsby.ucl.ac.uk

## Abstract

We introduce a new approach to learning in hierarchical latent-variable generative models called the "distributed distributional code Helmholtz machine", which emphasises flexibility and accuracy in the inferential process. Like the original Helmholtz machine and later variational autoencoder algorithms (but unlike adversarial methods) our approach learns an explicit inference or "recognition" model to approximate the posterior distribution over the latent variables. Unlike these earlier methods, it employs a posterior representation that is not limited to a narrow tractable parametrised form (nor is it represented by samples). To train the generative and recognition models we develop an extended wake-sleep algorithm inspired by the original Helmholtz machine. This makes it possible to learn hierarchical latent models with both discrete and continuous variables, where an accurate posterior representation is essential. We demonstrate that the new algorithm outperforms current state-of-the-art methods on synthetic, natural image patch and the MNIST data sets.

## 1   Introduction

There is substantial interest in applying variational methods to learn complex latent-variable generative models, for which the full likelihood function (after marginalising over the latent variables) and its gradients are intractable. Unsupervised learning of such models has two complementary goals: to learn a good approximation to the distribution of the observations; and also to learn the underlying structural dependence so that the values of latent variables may be inferred from new observations.

Variational methods rely on optimising a lower bound to the log-likelihood (the *free energy*), which depends on an approximation to the posterior distribution over the latents [1]. The performance of variational algorithms depends critically on the flexibility of the variational posterior. In cases where the approximating class does not contain the true posterior distribution, variational learning may introduce substantial bias to estimates of model parameters [2].

Variational autoencoders [3, 4] combine the variational inference framework with the earlier idea of the recognition network. This approach has made variational inference applicable to a large class of complex generative models. However, many challenges remain. Most current algorithms have difficulty learning hierarchical generative models with multiple layers of stochastic latent variables [5]. Arguably, this class of models is crucial for modelling data where the underlying physical process is itself hierarchical in nature. Furthermore, the generative models typically considered in the literature restrict the prior distribution to a simple form, most often a factorised Gaussian distribution, which makes it difficult to incorporate additional generative structure such as sparsity into the model.

We introduce a new approach to learning hierarchical generative models, the *Distributed Distributional Code (DDC) Helmholtz Machine*, which combines two ideas that originate in theoretical neuroscience: the Helmholtz Machine with wake-sleep learning [6]; and distributed or population codes for distributions [7, 8]. A key element of our method is that the approximate posterior distribution is represented as a set of expected sufficient statistics, rather than by directly parametrising the probability density function. This allows an accurate posterior approximation without being restricted to a rigid parametric class. At the same time, the DDC Helmholtz Machine retains some of the simplicity of the original Helmholtz Machine in that it does not require propagating gradients across different layers of latent variables. The result is a robust method able to learn the parameters of each layer of a hierarchical generative model with far greater accuracy than achieved by current variational methods.

We begin by briefly reviewing variational learning (Section 2), deep exponential family models (Section 3), and the original Helmholtz Machine and wake-sleep algorithm (Section 4); before introducing the DDC (Section 5.1), and associated Helmholtz Machine (Section 5.2), whose performance we evaluate in Section 7.

## 2 Variational inference and learning in latent variable models

Consider a generative model for observations $x$, that depend on latent variables $z$. Variational methods rely on optimising a lower bound on the log-likelihood by introducing an approximate posterior distribution $q(z|x)$ over the latent variables:

$$\log p_\theta(x) \geq \mathcal{F}(q, \theta, x) = \log p_\theta(x) - D_{KL}[q(z|x)||p_\theta(z|x)] \tag{1}$$

The cost of computing the posterior approximation for each observation can be efficiently amortised by using a recognition model [9], an explicit function (with parameters $\phi$, often a neural network) that for each $x$ returns the parameters of an estimated posterior distribution: $x \mapsto q_\phi(z)$.

A major source of bias in variational learning comes from the mismatch between the approximate and exact posterior distributions. The variational objective penalises this error using the 'exclusive' Kullback-Leibler divergence (see Eq. 1), which typically results in an approximation that underestimates the posterior uncertainty [10].

Multi-sample objectives (IWAE [11]; VIMCO [12]) have been proposed to remedy the disadvantages of a restrictive posterior approximation. Nonetheless, benefits of these methods are limited in cases when the proposal distribution is too far from the true posterior (see Section 7).

## 3 Deep exponential family models

We consider hierarchical generative models in which each conditional belongs to an exponential family, also known as *deep exponential family models* [13]. Let $x \in \mathcal{X}$ denote a single (vector) observation. The distribution of data $x$ is determined by a sequence of $L$ (vector) latent variables $z_1 \ldots z_L$ arranged in a conditional hierarchy as follows:

$$\boxed{z_L} \quad p(z_L) = \exp(\theta_L^T S_L(z_L) - \Phi_L(\theta_L))$$

$$\vdots$$

$$\boxed{z_2} \quad p(z_2|z_3) = \exp(g_2(z_3, \theta_2)^T S_2(z_2) - \Phi_2(g_2(z_3, \theta_2)))$$

$$\boxed{z_1} \quad p(z_1|z_2) = \exp\left(g_1(z_2, \theta_1)^T S_1(z_1) - \Phi_1(g_1(z_2, \theta_1))\right)$$

$$\boxed{x} \quad p(x|z_1) = \exp\left(g_0(z_1, \theta_0)^T S_0(x) - \Phi_0(g_0(z_1, \theta_0))\right)$$

Each conditional distribution is a member of a *tractable* exponential family, so that conditional sampling is possible. Using $l \in \{0, 1, 2, \ldots L\}$ to denote the layer (with $l = 0$ for the observation),

these distributions have sufficient statistic function $S_l$, natural parameter given by a known function $g_l$ of both the parent variable and a parameter vector $\theta_l$, and a log normaliser $\Phi_l$ that depends on this natural parameter. At the top layer, we lose no generality by taking $g_L(\theta_L) = \theta_L$.

We will maintain the general notation here, as the method we propose is very broadly applicable (both to continuous and discrete latent variables), provided that the family remains tractable in the sense that we can efficiently sample from the conditional distributions given the natural parameters.

## 4   The classic Helmholtz Machine and the wake-sleep algorithm

The Helmholtz Machine (HM) [6] comprises a latent-variable generative model that is to be fit to data, and a *recognition* network, trained to perform approximate inference over the latent variables.

The latent variables of an HM generative model are arranged hierarchically in a directed acyclic graph, with the variables in a given layer conditionally independent of one another given the variables in the layer above. In the original HM, all latent and observed variables were binary and formed a sigmoid belief network [14], which is a special case of deep exponential family models introduced in the previous section with $S_l(z_l) = z_l$ and $g_l(z_{l+1}, \theta_l) = \theta_l z_{l+1}$. The recognition network is a functional mapping with an analogous hierarchical architecture that takes each $x$ to an estimate of the posterior probability of each $z_l$, using a factorised mean-field representation.

The training of both generative model and recognition network follows a two-phase procedure known as *wake-sleep* [15]. In the *wake* phase, observations $x$ are fed through the recognition network to obtain the posterior approximation $q_\phi(z_l|x)$. In each layer the latent variables are sampled independently conditioned on the samples of the layer below according to the probabilities determined by the recognition model parameters. These samples are then used to update the generative parameters to increase the expected joint likelihood – equivalent to taking gradient steps to increase the variational free energy. In the *sleep* phase, the current generative model is used to provide joint samples of the latent variables and fictitious (or "dreamt") observations and these are used as supervised training data to adapt the recognition network. The algorithm allows for straightforward optimisation since parameter updates at each layer in both the generative and recognition models are based on locally generated samples of both the input and output of the layer.

Despite the resemblance to the two-phase process of expectation-maximisation and approximate variational methods, the sleep phase of wake-sleep does not necessarily increase the free-energy bound on the likelihood. Even in the limit of infinite samples, the mean field representation $q_\phi(z|x)$ is learnt so that it minimises $D_{KL}[p_\theta(z|x)\|q_\phi(z|x)]$, rather than $D_{KL}[q_\phi(z|x)\|p_\theta(z|x)]$ as required by variational learning. For this reason, the mean-field approximation provided by the recognition model is particularly limiting, since it not only biases the learnt generative model (as in the variational case) but it may also preclude convergence.

## 5   The DDC Helmholtz Machine (DDC-HM)

### 5.1   Distributed Distributional Codes

The key drawback of both the classic HM and most approximate variational methods is the need for a tractably parametrised posterior approximation. Our contribution is to instead adopt a flexible and powerful representation of uncertainty in terms of expected values of large set of (possibly random) arbitrary nonlinear functions. We call this representation a Distributed Distributional Code (DDC) in acknowledgement of its history in theoretical neuroscience [7, 8]. In the DDC-HM, each posterior is represented by approximate expectations of non-linear *encoding functions* $\{T^{(i)}(z)\}_{i=1...K}$ with respect to the *true posterior* $p_\theta(z|x)$:

$$r_l^{(i)}(x, \phi) \approx \left\langle T^{(i)}(z_l) \right\rangle_{p_\theta(z_l|x)}, \tag{2}$$

where $r_l^{(i)}(x, \phi), i = 1...K_l$ is the output of the recognition network (parametrised by $\phi$) at the $l$th latent layer, and the angle brackets denote expectations. A finite—albeit large—set of expectations does not itself fully specify the probability distribution $p_\theta(z|x)$. Thus, the recognition outputs $\{r_l^{(i)}\}_{i=1...K_l}$ are interpreted as representing an approximate posterior $q(z_l|x)$ defined by the distribution of maximum entropy that agrees with all of the encoded expectations.

A standard calculation shows that this distribution has a density of the form [1, Ch.3]:

$$q(z|x) = \frac{1}{Z(\eta(x))} \exp\left(\sum_{i=1}^{K} \eta^{(i)}(x) T^{(i)}(z)\right) \tag{3}$$

where the $\eta^{(i)}$ are natural parameters (derived as Lagrange multipliers enforcing the expectation constraints), and $Z(\eta)$ is a normalising constant. Thus, in this view, the encoded distribution $q$ is a member of the exponential family whose sufficient statistic functions correspond to the encoding functions $\{T^{(i)}(z)\}$, and the recognition network returns the expected sufficient statistics, or *mean parameters*. It follows that given a sufficiently large set of encoding functions, we can approximate the true posterior distribution arbitrarily well [16]. Throughout the paper we will use encoding functions of the following form:

$$T^{(i)}(z) = \sigma(w^{(i)T} z + b^{(i)}), \; i = 1 \ldots K \,, \tag{4}$$

where $w^{(i)}$ is a random linear projection with components sampled from a standard normal distribution, $b^{(i)}$ is a similarly distributed bias term, and $\sigma$ is a sigmoidal non-linearity. That is, the representation is designed to capture information about the posterior distribution along $K$ random projections in $z$-space. As a special case, we can recover the approximate posterior equivalent to the original HM if we consider linear encoding functions $T^{(i)}(z) = z_i$, corresponding to a factorised mean-field approximation.

Obtaining the posterior natural parameters $\{\eta^{(i)}\}$ (and thus evaluating the density in Eq. 3) from the mean parameters $\{r^{(i)}\}$ is not straightforward in the general case since $Z(\eta)$ is intractable. Thus, it is not immediately clear how a DDC representation can be used for learning. Our exact scheme will be developed below, but in essence it depends on the simple observation that most of the computations necessary for learning (and indeed most computations involving uncertainty) depend on the evaluation of appropriate expectations. Given a rich set of encoding functions $\{T^{(i)}\}_{i=1\ldots K}$ sufficient to approximate a desired function $f$ using linear weights $\{\alpha_i\}$, such expectations become easy to evaluate in the DDC representation:

$$f(z) \approx \sum_i \alpha^{(i)} T^{(i)}(z) \quad \Rightarrow \quad \langle f(z) \rangle_{q(z)} \approx \sum_i \alpha^{(i)} \left\langle T^{(i)}(z) \right\rangle_{q(z)} = \sum_i \alpha^{(i)} r^{(i)} \tag{5}$$

Thus, the richer the family of DDC encoding functions, the more accurate are both the approximated posterior distribution, *and* the approximated expectations[1]. We will make extensive use of this property in the following section where we discuss how this posterior representation is learnt (sleep phase) and how it can be used to update the generative model (wake phase).

## 5.2 The DDC Helmholtz Machine algorithm

---

**Algorithm 1** DDC Helmholtz Machine training

---

Initialise $\theta$
**repeat**
    **Sleep phase:**
    for $s = 1 \ldots S$, sample: $z_L^{(s)}, ..., z_1^{(s)}, x^{(s)} \sim p_\theta(x, z_1, ..., z_L)$
    update recognition parameters $\{\phi_l\}$ [eq. 7]
    update function approximators $\{\alpha_l, \beta_l\}$ [appendix]
    **Wake phase:**
    $x \leftarrow \{\text{minibatch}\}$
    evaluate $r_l(x, \phi)$ [eq. 8]
    update $\theta$: $\Delta\theta \propto \widehat{\nabla_\theta \mathcal{F}}(x, r(x, \phi), \theta)$ [appendix]
**until** $|\widehat{\nabla_\theta \mathcal{F}}| < \text{threshold}$

---

Following [6] the generative and recognition models in the DDC-HM are learnt in two separate phases (see Algorithm 1). The sleep phase involves learning a recognition network that takes data

points $x$ as input and produces expectations of the non-linear encoding functions $\{T^{(i)}\}$ as given by Eq. (2); *and* learning how to use these expectations to update the generative model parameters using approximations of the form of Eq. (5). The wake phase updates the generative parameters by computing the approximate gradient of the free energy, using the posterior expectations learned in the sleep phase. Below we describe the two phases of the algorithm in more detail.

**Sleep phase**  One aim of the sleep phase, given a current generative model $p_\theta(x, z)$, is to update the recognition network so that the Kullback-Leibler divergence between the true and the approximate posterior is minimised:

$$\phi = \operatorname{argmin} \mathrm{D}_{\mathrm{KL}}[p_\theta(z|x)||q_\phi(z|x)] \tag{6}$$

Since the DDC $q(z|x)$ is in the exponential family, the KL-divergence in Eq. (6) is minimised if the expectations of the sufficient statistics vector $\mathcal{T} = [T^{(1)}, \ldots, T^{(K)}]$ under the two distributions agree: $\langle \mathcal{T}(z) \rangle_{p_\theta(z|x)} = \langle \mathcal{T}(z) \rangle_{q_\phi(z|x)}$. Hence the parameters of the recognition model should be updated so that: $r_l(x, \phi_l) \approx \langle \mathcal{T}(z_l) \rangle_{p_\theta(z_l|x)}$ This requirement can be translated into an optimisation problem by sampling $z_L^{(s)}, \ldots, z_1^{(s)}, x^{(s)}$ from the generative model and minimising the error between the output of the recognition model $r_l(x^{(s)}, \phi_l)$ and encoding functions $\mathcal{T}$ evaluated at the generated *sleep* samples. For tractability, we substitute the squared loss in place of Eq. (6).

$$\phi_l = \operatorname{argmin} \sum_s \left\| r_l(x^{(s)}, \phi_l) - \mathcal{T}(z_l^{(s)}) \right\|^2 \tag{7}$$

In principle, one could use any function approximator (such as a neural network) for the recognition model $r_l(x^{(s)}, \phi_l)$, provided that it is sufficiently flexible to capture the mapping from the data to the encoded expectations. Here, we parallel the original HM, and use a recognition model that reflects the hierarchical structure of the generative model. For a model with 2 layers of latent variables:

$$h_1(x, W) = [Wx]_+, \qquad r_1(x, \phi_1) = \phi_1 \cdot h_1(W, x), \qquad r_2(x, \phi_2) = \phi_2 \cdot r_1(x, \phi_1), \tag{8}$$

where $W, \phi_1, \phi_2$ are matrices and $[.]_+$ is a rectifying non-linearity. Throughout this paper we use a fixed $W \in \mathsf{R}^{M \times D_x}$ sampled from a normal distribution, and learn $\phi_1, \phi_2$ according to Eq. (7).

Recognition model learning in the DDC-HM thus parallels that of the original HM, albeit with a much richer posterior representation. The second aim of the DDC-HM sleep phase is quite different: a further set of weights must be learnt to approximate the gradients of the generative model joint likelihood. This step is derived in the appendix, but summarised in the following section.

**Wake phase**  The aim in the wake phase is to update the generative parameters to increase the variational free energy $\mathcal{F}(q, \theta)$, evaluated on data $x$, using a gradient step:

$$\Delta\theta \propto \nabla_\theta \mathcal{F}(q, \theta) = \langle \nabla_\theta \log p_\theta(x, z) \rangle_{q(z|x)} \tag{9}$$

The update depends on the evaluation of an expectation over $q(z|x)$. As discussed in Section 5.1, the DDC approximate posterior representation allows us to evaluate such expectations by approximating the relevant functions using the non-linear encoding functions $\mathcal{T}$.

For deep exponential family generative models, the gradients of the free energy take the following form (see appendix):

$$\nabla_{\theta_0}\mathcal{F} = \nabla_{\theta_0} \langle \log p(x|z_1) \rangle_q = S_0(x)^T \langle \nabla g(z_1, \theta_0) \rangle_{q(z_1)} - \langle \mu_{x|z_1}^T \nabla g(z_1, \theta_0) \rangle_{q(z_1)}$$
$$\nabla_{\theta_l}\mathcal{F} = \nabla_{\theta_l} \langle \log p(z_l|z_{l+1}) \rangle_q = \langle S_l(z_l)^T \nabla g(z_{l+1}, \theta_l) \rangle_{q(z_l, z_{l+1})} - \langle \mu_{z_l|z_{l+1}}^T \nabla g(z_{l+1}, \theta_l) \rangle_{q(z_{l+1})}$$
$$\nabla_{\theta_L}\mathcal{F} = \nabla_{\theta_L} \langle \log p(z_L) \rangle_q = \langle S_L(z_L) \rangle - \nabla\Phi(\theta_L), \tag{10}$$

where $\mu_{x|z_1}, \mu_{z_l|z_{l+1}}$ are expected sufficient statistic vectors of the conditional distributions from the generative model: $\mu_{x|z_1} = \langle S_0(x) \rangle_{p(x|z_1)}, \mu_{z_l|z_{l+1}} = \langle S_l(z_l) \rangle_{p(z_l|z_{l+1})}$. Now the functions that must be approximated are the functions of $\{z_l\}$ that appear within the expectations in Eqs. 10. As shown in the appendix, the coefficients of these combinations can be learnt by minimising a squared error on the sleep-phase samples, in parallel with the learning of the recognition model.

Thus, taking the gradient in the first line of Eq. (10) as an example, we write $\nabla_\theta g(z_1, \theta_0) \approx \sum_i \alpha_0^{(i)} T^{(i)}(z_1) = \alpha_0 \cdot \mathcal{T}(z_l)$ and evaluate the gradients as follows:

$$\text{sleep:} \qquad \alpha_0 \leftarrow \operatorname{argmin} \sum_s \left( \nabla_\theta g(z_1^{(s)}, \theta_0) - \alpha_0 \cdot \mathcal{T}(z_1^{(s)}) \right)^2 \qquad (11)$$

$$\text{wake:} \qquad \langle \nabla_\theta g(z_1, \theta_0) \rangle_{q(z_1)} \approx \alpha_0 \cdot \langle \mathcal{T}(z_1) \rangle_{q(z_1)} = \alpha_0 \cdot r_1(x, \phi_1) \qquad (12)$$

with similar expressions providing all the gradients necessary for learning derived in the appendix.

In summary, in the DDC-HM computing the wake-phase gradients of the free energy becomes straightforward, since the necessary expectations are computed using approximations learnt in the sleep phase, rather than by an explicit construction of the intractable posterior. Furthermore, as shown in the appendix, using the function approximations trained using the sleep samples and the posterior representation produced by the recognition network, we can learn the generative model parameters without needing any explicit independence assumptions (within or across layers) about the posterior distribution.

## 6 Related work

Following the Variational Autoencoder (VAE; [3, 4]), there has been a renewed interest in using recognition models – originally introduced in the HM – in the context of learning complex generative models. The Importance Weighted Autoencoder (IWAE; [11]) optimises a tighter lower bound constructed by an importance sampled estimator of the log-likelihood using the recognition model as a proposal distribution. This approach decreases the variational bias introduced by the factorised posterior approximation of the standard VAE. VIMCO [12] extends this approach to discrete latent variables and yields state-of-the-art generative performance on learning sigmoid belief networks. We compare our method to the IWAE and VIMCO in section 7. Sonderby et al. [5] demonstrate that the standard VAE has difficulty making use of multiple stochastic layers. To overcome this, they propose the Ladder Variational Autoencoder with a modified parametrisation of the recognition model that includes stochastic top-down pass through the generative model. The resulting posterior approximation is a factorised Gaussian as for the VAE. Normalising Flows [18] relax the factorised Gaussian assumption on the variational posterior. Through a series of invertible transformations, an arbitrarily complex posterior can be constructed. However, to our knowledge, they have not yet been successfully applied to deep hierarchical generative models.

## 7 Experiments

We have evaluated the performance of the DDC-HM on a directed graphical model comprising two stochastic latent layers and an observation layer. The prior on the top layer is a mixture of Gaussians, while the conditional distributions linking the layers below are Laplace and Gaussian, respectively:

$$p(z_2) = 1/2 \left( \mathcal{N}(z_2|m, \sigma^2) + \mathcal{N}(z_2|-m, \sigma^2) \right)$$
$$p(z_1|z_2) = \text{Laplace}(z_1|\mu = 0, \lambda = \text{softplus}(Bz_2)) \qquad (13)$$
$$p(x|z_1) = \mathcal{N}(x|\mu = \Lambda z_1, \Sigma_x = \Psi_{diag})$$

We chose a generative model with a non-Gaussian prior distribution and sparse latent variables, models typically not considered in the VAE literature. Due to the sparsity and non-Gaussianity, learning in these models is challenging, and the use of a flexible posterior approximation is crucial. We show that the DDC-HM can provide a sufficiently rich posterior representation to learn accurately in such a model. We begin with low dimensional synthetic data to evaluate the performance of the approach, before evaluating performance on a data set of natural image patches.

**Synthetic examples** To illustrate that the recognition network of the DDC-HM is powerful enough to capture dependencies implied by the generative model, we trained it on a data set generated from the model (N=10000). The dimensionality of the observation layer, the first and second latent layers was set to $D_x = 2, D_1 = 2, D_2 = 1$, respectively, for both the true generative model and the fitted models. We used a recognition model with a hidden layer of size 100, and $K_1 = K_2 = 100$ encoding functions for each latent layer, with 200 sleep samples, and learned the parameters of the conditional distributions $p(x|z_1)$ and $p(z_1|z_2)$ while keeping the prior on $z_2$ fixed (m=3, $\sigma$=0.1).

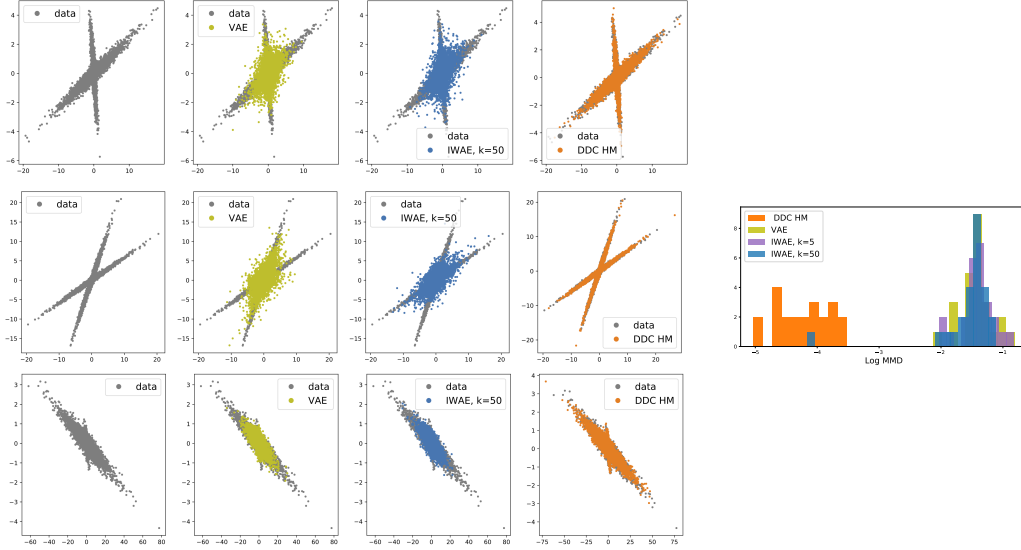

Figure 1: Left: Examples of the distributions learned by the Variational Autoencoder (VAE), the Importance Weighted Variational Autoencoder (IWAE) with k=50 importance samples and the DDC Helmholtz Machine. Right: histogram of $\log$ MMD values for different algorithms trained on synthetic datasets.

As a comparison, we have also fitted both a Variational Autoencoder (VAE) and an Importance Weighted Autoencoder (IWAE), using 2-layer recognition networks with 100 hidden units each, producing a factorised Gaussian posterior approximation (or proposal distribution for the IWAE). To make the comparison between the algorithms clear (i.e. independent of initial conditions, local optima of the objective functions) we initialised each model to the true generative parameters and ran the algorithms until convergence (1000 epochs, learning rate: $10^{-4}$, using the Adam optimiser; [19]).

Figure 1 shows examples of the training data and data generated by the VAE, IWAE and DDC-HM models after learning. The solution found by the DDC-HM matches the training data, suggesting that the posterior approximation was sufficiently accurate to avoid bias during learning. The VAE, as expected from its more restrictive posterior approximation, could capture neither the strong anti-correlation between latent variables nor the heavy tails of the distribution. Similar qualitative features are seen in the IWAE samples, suggesting that the importance weighting was unable to recover from the strongly biased posterior proposal.

We quantified the quality of the fits by computing the *maximum mean discrepancy* (MMD) [17] between the training data and the samples generated by each model [20] [2]. We used an exponentiated quadratic kernel with kernel width optimised for maximum test power [21]. We computed the MMD for 25 data sets drawn using different generative parameters, and found that the MMD estimates were significantly lower for the DDC-HM than for the VAE or the IWAE (k=5, 50) (Figure 1).

Beyond capturing the density of the data, correctly identifying the underlying latent structure is also an important criterion when evaluating algorithms for learning generative models. Figure 2 shows an example where we have used Hamiltonian Monte Carlo to generate samples from the true posterior distribution for one data point under the generative models learnt by each approach. We found that there was close agreement between the posterior distributions of the true generative model and the one learned by the DDC-HM. However, the biased recognition of the VAE and IWAE in turn biases the learnt generative parameters so that the resulting posteriors (even when computed without the recognition networks) appear closer to Gaussian.

**Natural image patches**    We tested the scalability of the DDC-HM by applying it to a natural image data set [22]. We trained the same generative model as before on image patches with dimensionality

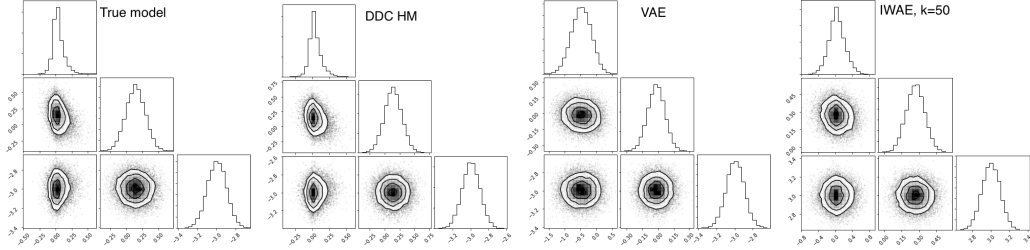

Figure 2: Example posteriors corresponding to the learned generative models. The corner plots show the pairwise and marginal densities of the three latent variables, for the true model (top left), the model learned by the VAE, IWAE (k=50) and DDC-HM.

$D_x = 16 \times 16$ and varying sizes of latent layers. The recognition model had a hidden layer of size 500, $K_1 = 500$, $K_2 = 100$ encoding functions for $z_1$ and $z_2$, respectively, and used 1000 samples during the sleep phase. We compared the performance of our model with the IWAE (k=50) using the relative (three sample) MMD test [20] with a exponentiated quadratic kernel (width chosen by the median heuristic). The test establishes whether the MMD distance between distributions $P_x$ and $P_y$ is significantly smaller than the distance between $P_x$ and $P_z$. We used the image data set as our reference distribution and the IWAE being closer to the data as null hypothesis. Table 1 summarises the results obtained on models with different latent dimensionality, all of them strongly preferring the DDC-HM.

Table 1: 3-sample MMD results. The table shows the results of the 'relative' MMD test between the DDC-HM and the IWAE ($k = 50$) on the image patch data set for different generative model architectures. The null hypothesis tested: $\text{MMD}_{\text{IWAE}} < \text{MMD}_{\text{DDC-HM}}$. Small p values indicate that the model learned by the DDC HM matches the data significantly better than the one learned by the IWAE ($k = 50$). We obtained similar results when comparing to IWAE $k = 1, 5$ (not shown).

| LATENT DIMENSIONS | | IWAE | DDC HM | p-value |
|---|---|---|---|---|
| $D_1 = 10$ | $D_2 = 10$ | 0.126 | 0.0388 | $\ll 10^{-5}$ |
| $D_1 = 50$ | $D_2 = 2$ | 0.0754 | 0.0269 | $\ll 10^{-5}$ |
| $D_1 = 50$ | $D_2 = 10$ | 0.247 | 0.00313 | $\ll 10^{-5}$ |
| $D_1 = 100$ | $D_2 = 2$ | 0.076 | 0.0211 | $\ll 10^{-5}$ |
| $D_1 = 100$ | $D_2 = 10$ | 0.171 | 0.00355 | $\ll 10^{-5}$ |

**Sigmoid Belief Network trained on MNIST**    Finally, we evaluated the capacity of our model to learn hierarchical generative models with discrete latent variables by training a sigmoid belief network (SBN). We used the binarised MNIST dataset of 28x28 images of handwritten digits [23]. The generative model had three layers of binary latent variables, with dimensionality of 200 in each layer. The recognition model had a sigmoidal hidden layer of size 300 and DDC representations of size 200 for each latent layer. As a comparison, we have also trained an SBN with the same architecture using the VIMCO algorithm (as described in [12]) with 50 samples from the proposal distribution [3]. To quantify the fits, we have performed the relative MMD test using the test set ($N = 10000$) as a reference distribution and two sets of samples of the same size generated from the SBN trained by the VIMCO and DDC-HM algorithms. Again, we used an exponentiated quadratic kernel with width chosen by the median heuristic. The test strongly favoured the DDC-HM over VIMCO with $p \ll 10^{-5}$ (with MMD values of $6 \times 10^{-4}$ and $2 \times 10^{-3}$, respectively).

## 8   Discussion

The DDC Helmholtz Machine offers a novel approach to learning hierarchical generative models, which combines the basic idea of the wake-sleep algorithm with a flexible posterior representation.

The lack of strong parametric assumptions in the DDC representation allows the algorithm to learn generative models with complex posterior distributions accurately.

As in the classical Helmholtz Machine, the approximate posterior is found by seeking to minimise the "reverse" divergence $D_{\mathrm{KL}}[p(z|x)\|q(z|x)]$, albeit within a much richer class of distributions. Thus, the modified wake-sleep algorithm presented here still does not directly optimise a variational lower bound on the log-likelihood. Rather, it can be viewed as following an approximation to the *gradient* of the log-likelihood, where the quality of the approximation depends on the richness of the DDC representation used. Precise conditions for convergence are yet to be established, but the expectation is that when the approximation is rich enough for the error in the resulting gradient estimate to be bounded, the algorithm will always reach a region around a local mode in which the true gradient does not exceed that error bound.

The DDC-HM recognition model can be trained layer-by-layer using the samples from the generative model, with no need to back-propagate gradients across stochastic layers. In the version discussed here, the recognition network depended on linear mappings between encoding functions and a fixed non-linear basis expansion of the input. This restrictive form allowed for closed-form updates in the sleep phase. However, this assumption could be relaxed by introducing a neural network between each latent variable layer, along with a modified learning scheme in the sleep phase. This approach may increase the accuracy of the posterior expectations computed during the wake phase.

Another future direction involves learning the non-linear encoding functions or choosing them in accordance with the properties of the generative model (e.g. requiring sparsity in the random projections). Finally, a natural extension of the DDC representation with expectations of a finite number of encoding functions, would be to approach the RKHS mean embedding, corresponding to infinitely many encoding functions [24, 25].

Even without these extensions, however, the DDC-HM offers a novel and powerful approach to probabilistic learning with complex hierarchical models.

### Acknowledgments

This work was funded by the Gatsby Charitable Foundation.

## Footnotes

[1]In a suitable limit, an infinite family of encoding functions would correspond to a mean embedding representation in a reproducing kernel Hilbert space [17]

[2]Estimating the log-likelihood by importance sampling in this model has proven to be unreliable due the lack of a good proposal distribution

[3]The model achieved an estimated negative log-likelihood of 90.97 nats, similar to the one reported by [12] (90.9 nats)

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
