[Supplementary Material · supplementary.pdf]

# Appendix

**Computing and learning gradients for the generative model parameters**

The variational free energy for a hierarchical generative model over observations $x$ with latent variables $z_1 \ldots z_L$ can be written:

$$\mathcal{F}(q, \theta) = \langle \log p(x|z_1) \rangle_{q(z_1)} + \sum_{l=1}^{L-1} \langle \log p(z_l|z_{l+1}) \rangle_{q(z_l, z_{l+1})} + \langle \log p(z_L) \rangle_{q(z_L)} + H[q(z_1 \ldots z_L)] \tag{1}$$

where the distributions $q$ represent components of the approximate posterior and $H[\cdot]$ is the Shannon entropy. We take each conditional distribution to have exponential family form. Thus (for example)

$$\log p(z_l|z_{l+1}) = g_l(z_{l+1}, \theta_l)^T S_l(z_l) - \Phi_l(g_l(z_{l+1}, \theta_l)) \tag{2}$$

from which it follows that:

$$\nabla_{\theta_l} \log p(z_l|z_{l+1}) = S_l(z_l)^T \nabla_{\theta_l} g_l(z_{l+1}, \theta_l) - \Phi_l'(g_l(z_{l+1}, \theta_l)) \nabla_{\theta_l} g_l(z_{l+1}, \theta_l) \tag{3}$$

$$= S_l(z_l)^T \nabla_{\theta_l} g_l(z_{l+1}, \theta_l) - \mu_{z_1|z_{l+1}} \nabla_{\theta_l} g_l(z_{l+1}, \theta_l) \tag{4}$$

where we have used the standard result that the derivative of the log normaliser of an exponential family distribution with respect to the natural parameter is the mean parameter

$$\mu_{z_l|z_{l+1}} = \langle S_l(z_l) \rangle_{p_{\theta_l}(z_l|z_{l+1})}. \tag{5}$$

Thus, it follows that:

$$\nabla_{\theta_0} \mathcal{F} = \nabla_{\theta_0} \langle \log p(x|z_1) \rangle_{q(z_1)} = S_0(x)^T \langle \nabla g(z_1, \theta_0) \rangle_{q(z_1)} - \langle \mu_{x|z_1}^T \nabla g(z_1, \theta_0) \rangle_{q(z_1)} \tag{6}$$

$$\nabla_{\theta_l} \mathcal{F} = \nabla_{\theta_l} \langle \log p(z_l|z_{l+1}) \rangle_q = \langle S_l(z_l)^T \nabla g(z_{l+1}, \theta_l) \rangle_{q(z_l, z_{l+1})} - \langle \mu_{z_l|z_{l+1}}^T \nabla g(z_{l+1}, \theta_l) \rangle_{q(z_{l+1})} \tag{7}$$

(for $l = 1 \ldots L - 1$) and

$$\nabla_{\theta_L} \mathcal{F} = \nabla_{\theta_L} \langle \log p(z_L) \rangle_q = \langle S_L(z_L) \rangle - \nabla \Phi(\theta_L) \tag{8}$$

These are the gradients that must be computed in the wake phase.

In order to compute these gradients from the DDC posterior representation we need to express the functions of $z_l$ that appear in Eqs. 6–8 as linear combinations of the encoding functions $T(z_l)$. The linear coefficients can be learnt using samples from the generative model produced during the sleep phase. The example given in the main paper is the most straightforward. We wish to find

$$\nabla_{\theta_0} g(z_1, \theta_0) \approx \sum_i \alpha_0^{(i)} T^{(i)}(z_1) \tag{9}$$

in which the coefficients $\alpha_0$ can be obtained from samples by evaluating the gradient of $g$ with respect to $\theta_0$ at $z_1^{(s)}$ and minimising the squared error:

$$\alpha_0 \leftarrow \arg\min \sum_s \left( \nabla_\theta g(z_1^{(s)}, \theta_0) - \alpha_0 \cdot \mathcal{T}(z_1^{(s)}) \right)^2. \tag{10}$$

Once these coefficients have been found in the sleep phase, the wake phase expectations are found from the DDC recognition model very simply:

$$\langle \nabla_{\theta_0} g(z_1, \theta_0) \rangle_{q(z_1)} \approx \sum_i \alpha_0^{(i)} \langle T^{(i)}(z_1) \rangle_{q(z_1)} \tag{11}$$

$$= \sum_i \alpha_0^{(i)} r_1^{(i)}(x, \phi_1) \tag{12}$$

Some of the gradients (see Eq.7) require taking expectations using the joint posterior distribution $q(z_l, z_{l+1}|x)$. However, the recognition network as we have described it in the main paper only contains information about the marginal posteriors $q(z_l|x), q(z_{l+1}|x)$. It turns out that it is nevertheless possible to estimate these expectations without imposing an assumption that the approximate posterior factorises across the layers $z_l, z_{l+1}$.

We begin by noticing that due to the structure of the generative model the posterior distribution $q$ can be factorised into $q(z_{l+1}, z_l|x) = p(z_{l+1}|z_l)q(z_l|x)$ without any further assumptions (where $p(z_{l+1}|z_l)$) is the conditional implied by the generative model. Thus, we can rewrite the term in Equation 7 as:

$$\langle S_l(z_l)^T \langle \nabla_{\theta_l} g(z_{l+1}, \theta_l) \rangle_{p(z_{l+1}|z_l)} \rangle_{q(z_l)}$$

Now, we can replace the expectation $\langle \nabla_{\theta_l} g(z_{l+1}, \theta_l) \rangle_{p(z_{l+1}|z_l)}$ by an estimate using *sleep* samples, i.e. for each pair of samples $\{z_l^{(s)}, z_{l+1}^{(s)}\}$ from the prior, we have a single sample from the true posterior distribution $z_{l+1}^{(s)} \sim p(z_{l+1}|z_l = z_l^{(s)})$. Thus, we obtain coefficients $\alpha_l$ during the sleep phase so:

$$\alpha_l = \operatorname{argmin} \sum_s \left( S_l(z_l^{(s)})^T \nabla_{\theta_l} g(z_{l+1}^{(s)}, \theta_l) - \alpha_l \cdot \mathcal{T}(z_l^{(s)}) \right)^2 \tag{13}$$

which will converge so that

$$\alpha_l \cdot \mathcal{T}(z_l) \approx S_l(z_l)^T \langle \nabla_{\theta_l} g(z_{l+1}, \theta_l) \rangle_{p(z_{l+1}|z_l)}. \tag{14}$$

Now, during the wake phase it follows that

$$\langle S_l(z_l)^T \nabla_{\theta_l} g(z_{l+1}, \theta_l) \rangle_{q(z_{l+1}, z_l)} = \langle S_l(z_l)^T \langle \nabla_{\theta_l} g(z_{l+1}, \theta_l) \rangle_{p(z_{l+1}|z_l)} \rangle_{q(z_l)} \tag{15}$$

$$\approx \left\langle \sum_i \alpha_l^{(i)} T^{(i)}(z_l) \right\rangle_{q(z_l)} \tag{16}$$

$$= \sum_i \alpha_l^{(i)} r_l^{(i)}(x, \phi_l) \tag{17}$$

Thus, all the function approximations needed to evaluate the gradients are carried out by using samples from the generative model (*sleep* samples) as training data. The full set of necessary function approximations with (matrix) parameters describing the linear mappings denoted by $\{\alpha_l\}_{l=0...L}, \{\beta_l\}_{l=1...L}$ is:

$$\alpha_0 : \mathcal{T}(z_1^{(s)}) \mapsto \nabla_\theta g(z_1^{(s)}, \theta_0) \tag{18}$$

$$\beta_l : \mathcal{T}(z_l^{(s)}) \mapsto \mu_{z_{l-1}|z_l^{(s)}}^T \nabla_\theta g(z_l^{(s)}, \theta_{l-1}) \tag{19}$$

$$\alpha_l : \mathcal{T}(z_l^{(s)}) \mapsto S_l(z_l^{(s)}) \nabla_\theta g(z_{l+1}^{(s)}, \theta_l) \tag{20}$$

$$\alpha_L : \mathcal{T}(z_L^{(s)}) \mapsto S_L(z_L^{(s)}) \tag{21}$$

where the expressions on the right hand side are easy to compute given the current parameters of the generative model. Note that $\mu_{z_{l-1}|z_l^{(s)}}$ appearing in equation 19 are expectations that can be evaluated analytically for tractable exponential family models, as a consequence of the conditionally independent structure of the generative model. Alternatively, they can also be estimated using the *sleep* samples, by training

$$\beta_l : \mathcal{T}(z_l^{(s)}) \mapsto S_{l-1}(z_{l-1}^{(s)})^T \nabla_\theta g(z_l^{(s)}, \theta_{l-1}) \tag{22}$$

Finally, putting the sleep and the wake phase together, the updates for the generative parameters during the wake phase are:

$$\begin{aligned} \Delta\theta_0 &\propto S_0(x)^T \alpha_0 r_1(x, \phi_1) - \beta_1 r_1(x, \phi_1) \\ \Delta\theta_l &\propto \alpha_l r_l(x, \phi_l) - \beta_{l+1} r_{l+1}(x, \phi_{l+1}) \\ \Delta\theta_L &\propto \alpha_L r_L(x, \phi_L) - \nabla_{\theta_L} \Phi(\theta_L) \end{aligned} \tag{23}$$