[Reviews · NeurIPS 2018]

Reviewer 1



This paper presents an alternative to variational autoencoders and other generative models with latent variables that rely on the wake-sleep algorithm for training. The main problem with the wake-sleep algorithm is its bias: the recognition model has different conditional independencies than the generative model and it's trained to optimize a different objective. DDC-HM solves this by instead working with sufficient statistics, and using those to implicitly define the maximum entropy distribution consistent with those statistics. The measurements chosen are random functions. The methods are evaluated on synthetic data and two small vision datasets (image patches and MNIST), comparing against two baselines using the MMD metric. The idea sounds promising. I don't know the related work well enough to evaluate the novelty with confidence. The evidence in support of the method is adequate. If this paper were to be revised or expanded, I would be interested to know if there are any convergence guarantees (perhaps given certain assumptions or conditions). The bias of HMs was listed as one of the motivations for developing DDC-HM, but the biases of DDC-HMs are not analyzed theoretically or empirically -- just overall performance on a few datasets. A discussion of efficiency would also be welcome. Can DDC-HMs be applied to datasets with many more variables and data points? How do their scalability compare to other methods? With probabilistic inference, there's often a tradeoff between efficiency and accuracy. Since the claim is that DDC-HMs are "flexible and accurate", it would be good to quantify the cost of that flexibility and accuracy. One other curiosity: in the experiments on synthetic data, the recognition model has many more parameters than the generative model. Presumably, this is because the inverse function is hard to describe analytically, and thus requires more hidden units to approximate well. Is it normally the case that the recognition model should be so much more complex, relative to the generative mdoel? Is the greater complexity of the recognition model balanced by the fact that it can be trained on many generated samples, and is thus less likely to overfit? It's not essential, but I'd be interested in some discussion about this. Overall, this paper proposes an interesting way to make deep generative models more flexible and less biased, and shows promising empirical results. More theoretical and empirical analysis could make it stronger, but I think this paper is acceptable for publication as-is. Update after the author response: As before, I think the paper is ok as-is, although my confidence is only moderate. I hope the authors add more analysis, as time and page limits allow.

Reviewer 2



The authors describe a novel method for learning deep generative models using variational methods that replaces typical mean field approximations in the wake-sleep algorithm with more sophisticated approximations. In particular, they suggest an approach in which the posterior distributions are approximated by expectations with respect to a predetermined set of sufficient statistics: A recognition network is trained to output these posterior distributions for each sample. The approximate posteriors are then selected to be those distributions of maximum entropy that agree with these expectations. As MLE of probabilisitc models can be reduced to computing expectations (i.e., the computation of the gradients), the entire procedure can be made tractable without the need of having to actually form the posterior distributions (no partition function computation necessary). For the most part, the paper is well-written and easy to follow. General comments: How are the sufficient statistics (i.e., the T's) selected in the experiments? There seem like there are a lot of knobs to turn here: choice of the form of the probability distributions, choice of the sufficient statistics, number of layers in the deep gen. network, etc. Can the authors comment a bit about how to go about this in practice? I wouldn't expect that this method (or any of the other methods for that matter) produces the same estimate of the posterior distributions on each run. Maybe it makes more sense to report average values? Or are you reporting the best fits as selected using a validation set? In either case, this needs to be made more clear if the experiments are to be replicated.

Reviewer 3



The paper describes a novel approach to perform posterior inference and learning in deep generative models. This work is mainly motivated by the fact that recent VAE approaches still have difficulties in learning models with multiple layers, and that they have difficulties with adequately capturing structured posteriors. The key contributions in this paper are * a novel approach to capture the posterior indirectly, using an inference network predicting expectations of random statistics w.r.t. the true posterior, and * demonstrating how to exploit this indirect representation for learning the generative model. To that end, the authors propose the Distributed Distributional Code (DDC) Helmholtz Machine, an extension to the classical Helmholtz Machine, including an extended wake-sleep algorithm. Strengths: + a quite refreshing approach to approximate inference and learning. The proposed approach combines several existing techniques in an intriguing way. In particular, I think that the approach to represent the posterior via estimated expectations of random statistics is a very powerful idea, with a potentially large impact on approximate inference/learning. Also the approach to circumvent an explicit representation of the approximate posterior density, but rather to derive required expectations (gradients) in a direct way, is a promising direction. + Experimental evaluations demonstrate -- to a certain extent -- the efficacy of the approach, with partially dramatic improvements over variational approaches. Weaknesses: - the approach inherits the disadvantages of wake-sleep. Thus, there is no clear objective optimized here and the method might even diverge. Furthermore, there is an additional approximation step which was not present in the original wake-sleep algorithm, namely the prediction of the required expectations/gradients. While the authors mention that the original wake-sleep algorithm has these drawbacks, they don't discuss them at all in their approach. - the motivation to fix VAEs' problem to learn deep models is addressed rather briefly in the experiments, by learning a model over binarized MNIST and 3 hidden sigmoid layers. This is not too convincing. Quality: The paper is technically sound, in that way that the choices made are clear and very convincing. However, as the approach is based in wake-sleep, it is clear that the approach is heuristic, in particular as it includes a further approximation (expectations/gradients used for learning). The consequences of this fact and weaknesses of the model should be discussed in a more open way and ideally be addressed in the experimental evaluation (e.g. showing fail cases, or discussing why there are none). Clarity: The paper is very well written and easy to follow. Originality: The approach presented is, to the best of my knowledge, novel, natural and elegant. Significance: Although the proposed is heuristic, it has a high potential to trigger further research in this direction. Summary: while the paper has its weaknesses (heuristic, which should be better discussed), its originality and potential inspiration for further work in this direction are the main reasons why I recommend an accept. *** EDIT *** I read the authors' reply and I'm rather happy with it. While their point is convincing that the approximation in DDC-HM can be made arbitrarily close, I'm still not sure if we can guarantee stable training. The authors claim that it can be shown that the algorithm approaches the vicinity of a stationary point of the likelihood, if the gradient approximation error is small. I think this might be a crucial point, but I don't fully understand if this implies convergences (rather than oscillatory behavior) and how small the error needs to get. In any case, the authors should include this result in their paper. And again, nice and refreshing work, I stick with my original rating.